# Genetic Variation, Polyploidy, Hybridization Influencing the Aroma Profiles of *Rosaceae* Family

**DOI:** 10.3390/genes15101339

**Published:** 2024-10-18

**Authors:** Xi Chen, Yu Zhang, Weihua Tang, Geng Zhang, Yuanhua Wang, Zhiming Yan

**Affiliations:** 1School of Agronomy and Horticulture, Jiangsu Vocational College of Agriculture and Forest, Jurong 212400, China; weihuatang@jsafc.edu.cn (W.T.); zhanggeng@jsafc.edu.cn (G.Z.); wangyuanhua@jsafc.edu.cn (Y.W.); yanzhim@jsafc.edu.cn (Z.Y.); 2Engineering and Technical Center for Modern Horticulture, Jurong 212400, China; 3Key Laboratory of Tobacco Biology and Processing, Ministry of Agriculture, Tobacco Research Institute, Chinese Academy of Agricultural Sciences (CAAS), Qingdao 266101, China; zhangyu02@caas.cn

**Keywords:** *Rosaceae*, polyploidy, aroma, fragrance, volatile organic compound, isoenzyme

## Abstract

Background: The fragrance and aroma of *Rosaceae* plants are complex traits influenced by a multitude of factors, with genetic variation standing out as a key determinant which is largely impacted by polyploidy. Polyploidy serves as a crucial evolutionary mechanism in plants, significantly boosting genetic diversity and fostering speciation. Objective: This review focuses on the *Rosaceae* family, emphasizing how polyploidy influences the production of volatile organic compounds (VOCs), which are essential for the aromatic characteristics of economically important fruits like strawberries, apples, and cherries. The review delves into the biochemical pathways responsible for VOC biosynthesis, particularly highlighting the roles of terpenoids, esters alcohols, aldehydes, ketones, phenolics, hydrocarbons, alongside the genetic mechanisms that regulate these pathways. Key enzymes, such as terpene synthases and alcohol acyltransferases, are central to this process. This review further explores how polyploidy and hybridization can lead to the development of novel metabolic pathways, contributing to greater phenotypic diversity and complexity in fruit aromas. It underscores the importance of gene dosage effects, isoenzyme diversity, and regulatory elements in determining VOC profiles. Conclusions: These findings provide valuable insights for breeding strategies aimed at improving fruit quality and aligning with consumer preferences. Present review not only elucidates the complex interplay between genomic evolution and fruit aroma but also offers a framework for future investigations in plant biology and agricultural innovation.

## 1. Polyploidy in Rosaceae Species Genome

Whole-genome duplication (WGD) making polyploidy is a pivotal mechanism in the evolution of plants, as noted by Soltis et al. [1] and Eric Schranz et al. [2]. Virtually all plants have undergone at least one episode of genome doubling [3], with approximately 35% of vascular plants being polyploid [4]. Numerous genomic studies have revealed ancient whole-genome duplication events [5]. Polyploidization also serves as a critical force in speciation. The mechanisms for the formation of polyploids primarily include unreduced gametes, somatic doubling, and hybridization. Polyploidy, with its enhanced genetic variation and diversity, effectively mitigates the detrimental effects of redundant genes. It facilitates the emergence of a broader array of functional traits through sub-functionalization and neo-functionalization [6,7] granting polyploids a competitive edge over diploids in environmental adaptability and allowing them to thrive in extreme conditions [8]. Certain redundant genes can escape the harmful consequences of redundancy by silencing their expression through methylation [6]. Additionally, polyploidy contributes to an increase in phenotypic diversity within species [9]. The *Rosaceae* family is a diverse and widespread group of plants known for encompassing a variety of well-known horticultural plants [10]. Key genera such as *Rosa*, *Malus*, *Pyrus*, *Prunus*, *Fragaria*, and *Rubus* form the backbone of the family [11]. The genomes of *Rosaceae* family plants exhibit the intriguing feature of having undergone complex genetic variation (including single nucleotide polymorphisms, insertions and deletions, copy number variation, structure variation) and multiple (auto- and allo-) polyploidy events [12]. The genome duplication (autopolyploidization) in the *Malus* genus (apple) is believed to have occurred around 50 million years ago [13], while *Fragaria viginiana* and *Fragaria chiloensis* (wild octoploid progenitors of *Fragaria* × *ananassa*) are from the merger of four diploid progenitor (allo-polyploidy) species approximately 1 million years ago [14].

## 2. *Rosaceae* Aromas: The Chemical and Genes

The fragrance of flowers and the aroma of fruits represent significant phenotypic diversity with considerable commercial potential in *Rosaceae* crops. Normally the volatile organic compounds (VOCs) contribute to flower and fruit aroma including terpenoid (like limonene, myrcene, and pinene), ester (like isoamyl acetate and ethyl butanoate giving banana and pineapple flavour respectively), alcohol (like hexanol in green apples), aldehyde (like benzaldehyde giving almond and cherry aroma), ketones (like diacetyl giving buttery flavour), phenolics (like vanillin, which provides a vanilla scent, and eugenol, which gives a clove-like aroma), lactones (like γ-decalactone giving peachy flavour) [15].

Take strawberry as example. The aroma of strawberry fruit is composed of a complex mixture of various volatile compounds. Reports indicate the presence of over 360 volatile aromatic compounds in strawberries, including esters, aldehydes, ketones, alcohols, alkenes, furanones, and sulfur-containing compounds [16]. Ester compounds are the most abundant constituents among the volatile compounds in strawberries, accounting for 25% to 90% of the total volatiles, and they impart the characteristic fruit aroma. The volatile esters produced by fresh strawberries primarily include ethyl esters, butyl esters, and hexyl esters, such as hexyl acetate, methyl butyrate, ethyl butyrate, methyl hexanoate, and ethyl hexanoate [17]. Furan compounds contribute to the distinctive aroma of strawberries, evoking a caramel scent. Notable representatives include 4-hydroxy-2,5-dimethyl-3(2H)-furanone (HDMF) and 4-methoxy-2,5-dimethyl-3(2H)-furanone (DMMF), which are unique volatile compounds found in fresh strawberries. Alkenes constitute the largest and most diverse group of secondary metabolites. In *Fragaria × ananassa*, the predominant terpenoid compounds are linalool and nerolidol, which have low odor thresholds and impart floral and citrus aromas [18]. Nineteen sulfur compounds have been identified in strawberries, including hydrogen sulfide, methanethiol, and various alkyl methanthio compounds. Among these, methanethiol and methyl thioacetate are the primary volatile sulfur compounds, with methanethiol possessing the highest aroma activity value.

Many of the genes involved in the biosynthesis and regulation of VOCs have been identified in *Rosaceae* species, which include economically important fruit crops like apples, pears, cherries, strawberries, and peaches. For instance, terpene synthase (TPS) is the core enzyme of the pathway of terpenoid biosynthesis. Terpenoids represent the largest class of compounds in plant secondary metabolism, synthesized through two distinct pathways: the methylerythritol phosphate (MEP) pathway occurrs in the plastids, and the mevalonate (MVA) pathway take places in the cytosol [19]. Both pathways generate isopentenyl diphosphate (IPP) and its isomer dimethylallyl diphosphate (DMAPP). Under the action of various isoprenyltransferases, IPP and DMAPP condense to form precursors for terpenoid biosynthesis.

In the MEP pathway, pyruvate and glyceraldehyde-3-phosphate serve as direct precursors, with MEP acting as a crucial intermediate; IPP and DMAPP are catalyzed by geranyl diphosphate (GPP) synthase to form the simple direct precursor GPP. Conversely, in the MVA pathway, acetyl-CoA serves as the precursor for synthesis, where two molecules of IPP and one molecule of DMAPP condense under the influence of farnesyl diphosphate synthase (FPP) to produce the sesquiterpenoid precursor farnesyl diphosphate (FPP). These precursor substances are subsequently catalyzed by terpene synthases (TPS), leading to the formation of various mono- and sesquiterpenes, which, as primary terpenoid scaffolds, undergo further modifications (hydroxylation, acylation, carbonylation, methylation, and redox reactions) to generate volatile terpenoid compounds. In apples and pears TPS have been linked to the production of terpenoid VOCs that contribute to the characteristic aromas of these fruits.

In strawberries (*Fragaria* spp.), genes encoding alcohol acyltransferase (AAT) enzymes play a significant role in ester formation during fruit ripening, a major component of strawberry aroma [20]. AAT is the rate-limiting enzyme in the synthesis of esters via the fatty acid pathway, with ester content dependent on substrate availability and the specificity of the AAT enzyme. Different AAT enzymes catalyze various substrate molecules (alcohols and acyl-CoA) to form distinct types of esters. Variations in AAT enzymes lead to differences in the types and quantities of esters among different strawberry cultivars [20,21].

In recent years, researchers have identified several *AAT* gene coding sequences from pineapple strawberries, wild strawberries, and Chilean strawberries (*F. chiloensis*), including *SAAT*, *FaAAT2*, *VAAT*, and *FcAAT1*. Enzyme activity analyses indicate that the FaAAT2 protein exhibits high catalytic activity towards C6-10 alcohols, particularly showing the highest activity with hexanol. SAAT shows a preference for long-chain acyl-CoA and aliphatic C4-9 alcohols as substrates, catalyzing the formation of butyric acid ethyl ester, hexanoic acid ethyl ester, and octanoic acid ethyl ester. Meanwhile, VAAT from wild strawberries primarily acts on short-chain alcohols and is closely related to the synthesis of cinnamic alcohol [22,23,24].

Currently, the research on the regulation of AAT genes is still in its early stages. Li et al. investigated the two wild strawberry (*Fragaria vesca*) varieties ‘Hawaii4′ and ‘Ruegen’ using weighted gene co-expression network analysis (WGCNA) and identified three transcription factors (ERF, AP2, and WRKY) closely associated with AAT expression during fruit ripening [25]. Notably, the ERF gene (*FveERF*), when transiently overexpressed in the fruit, can activate AAT expression, thereby promoting the accumulation of esters within the fruit, indicating its significant role in strawberry ester synthesis.

The phenylpropanoid pathway is a crucial metabolic route in plants that converts the amino acid phenylalanine into various important compounds. It begins with phenylalanine being transformed into cinnamic acid by the enzyme phenylalanine ammonia-lyase (PAL). Cinnamic acid is then converted into p-coumaric acid by cinnamate-4-hydroxylase (C4H), which is further processed into caffeic acid through the action of 4-coumarate-CoA ligase (4CL). Key products of this pathway include flavonoids, which help protect plants from UV radiation and attract pollinators, and lignin, which provides structural support to plant cell walls. Additionally, the pathway produces other phenolic compounds that play roles in plant defense and environmental interactions, making it vital for plant health and ecological relationships. This pathway produces various aromatic compounds, including flavonoids and phenolic compounds, which contribute to the scent and flavor profiles of fruits. In stone fruits such as cherries and peaches, genes involved in the phenylpropanoid pathway have been studied for their role in producing VOCs that contribute to both aroma and colour [26]. The production of VOCs responsible for fruit odour is a complex trait that can be affected by genetic variation and polyploidy.

## 3. Gene Dosage Effect Could Be One of the Major Reasons for Polyploidy or Copy Number Variation Altering Fruit Odour

With additional copies of gene in polyploid plants or by genetic variation, there can be changes in the quantity of enzymes involved in the biosynthesis of VOCs. This can lead to increased or decreased production of specific scent compounds. For instance, increasing copy number of *NUDX1-1a* (coding Nudix hydrolase) caused gene dosage effect on geraniol (terpenoid) content in rose petal [27].

## 4. Genetic Diversity Can Lead to Changes in the Regulation of Gene Expression

The regulation on gene expression normally depends on transcriptomic factors and *cis*-regulatory elements. The variation relevant with these two factors could make the gene expression differed, which would thus influence flavour. In strawberry, it has been found that the variation in the *cis-* regulatory elements regions of alcohol AAT coding gene could result in differential expression during fruit development, producing a dynamic and evolving aroma [28].

## 5. The Enzyme Isoforms Involved into VOCs Biosynthesis Are Playing a Key Role Determining or Altering Flower and/or Fruit Aroma Profile

As showed above polyploid facilitates the emergence of a broader array of functional traits through sub-functionalization and neo-functionalization. In *Rosaceae*, diversity of fragrance and aroma are pivotal agronomy trait and rely on the sub-functionalization and neo-functionalization of isoenzymes involved in VOCs generation largely. Here some representative isoforms generating varied VOCs in plants were showed in Figure 1.

### 5.1. Substrate Specificity Variation

Isoforms of enzymes may have slightly different peptide sequences because of the coding DNA mutation and leading to changes in the active site—the part of the enzyme where substrates bind. These changes can alter the enzyme’s specificity for different substrates. The mutation could be brought in by genome duplication or DNA replication error. Furthermore, the hybridization and crossing make the isoenzyme spreading. For example (Figure 1a), in *Rosa chinensis* phenylpropanoid pathway, different caffeic acid O-methyltransferase (RcOMT) isoforms rather have varied substrate specificity, which leads to different VOCs production [29]. Comparing with RcOMT3, RcOMT2 have broader substrate affinity, can catalyze methyl esterification of both caffeic acid and orcinol. As showed in Figure 1b, comparing with diploid strawberry (*F. vesca*), some of the octoploid (*Fragaria × ananassa*) have different AAT isoforms might utilize a broader range of acyl donors, leading to the synthesis of various esters [30]. AATs from *F. vesca* cannot catalyze the esterification of Pentanoyl-CoA with hexanol, and those from *Fragaria × ananassa* can. One the other hand, the presence of multiple enzyme isoforms allows the metabolic pathway to branch out, producing a wider variety of end products from a core chemical compound. The isoform might divert substrate specificity into different pathway, leading to the synthesis of novel VOCs. For example (Figure 1c), two members of BAHD family of acyltransferases acetyl-CoA:benzylalcohol O-acetyltransferase (BEAT) and benzoyl-CoA:benzyl alcohol benzoyl transferase (BEBT) could catalyze benzyl alcohol into benzyl acetate and benzyl benzoate respectively [29].

### 5.2. Enzyme Kinetics Alteration

Isoforms of enzymes can exhibit variations in their catalytic efficiency. Along with the alteration in catalytic efficiency, there could be the accumulation of intermediate products which are different VOCs and altering the aroma. For instance in *Rosa chinensis* (terpenoid biosynthesis pathway, Figure 1d), terpene synthase (TPS) isoform (+)-(3S)-linalool nerolidol synthase 1 (RcLIN-NERS1) catalyzes geranyl pyrophosphate (GPP) to produce only nerolidol. Another TPS isoform (RcLIN-NERS2) catalyzes GPP to produce nerolidol which is however less and with some byproducts as ß-myrcene, limonene and ocimenes [31]. In *Malus* × *domestica cv*. Golden Delicious, multiple isoforms of lipoxygenase (LOXs) are involved in the generation of volatile organic compounds (VOCs) that contribute to the fruit’s aroma. Among these isoforms, LOX1:Md:1a produces 13-hydroxyoctadecadienoic acid (13-HODE) as its primary product, with 9-hydroxyoctadecadienoic acid (9-HODE) as a byproduct. In contrast, the isoform LOX1:Md:1c exhibits the opposite behavior, with 9-HODE as the main product and 13-HODE as the byproduct [32].

## 6. The Merging of Genomes and/or Hybridization from Different Species or Varieties in Can Bring Together Novel Combinations of Alleles That May Interact in Unique Ways to Affect VOC Production

Interestingly, genome merger polyploidy or hybridization may facilitate the evolution of new metabolic pathways and even produce novel scent compounds not found in the parent plants or diploid relatives. For instance, in strawberries (*Fragaria* spp.), polyploidy has been associated with the more complexity of flavour profiles than the diploidy species [33]. Isoenzyme stacking is a remarkable reason for new VOCs emergence.

Isoenzyme copy number and polyploidy have a significant and interrelated relationship. In polyploid plants, gene duplication leads to multiple copies of isoenzymes, enhancing enzyme function diversity. This diversity allows for greater metabolic flexibility, enabling polyploid plants to fine-tune their metabolic processes and synthesize a wider range of metabolites, including VOCs responsible for aroma. Consequently, the increased number and/or the combination of isoenzymes in polyploid plants, such as apple (*Malus* × *domestica*), contributes to the production of new VOCs through various biosynthetic pathways, resulting in a more complex and diverse aroma profile [34].

## 7. Conclusions

The intricate relationship between genome and fruit aroma in *Rosaceae* species underscores the profound impact of genetic mechanisms on VOCs production. Beside genetic variation, polyploidy, through genome duplications and allopolyploid events, and hybridization lead to the emergence of novel and complex aroma profiles. The genetic complexities allow for greater metabolic flexibility, as evidenced by the presence of multiple isoenzymes that influence VOC biosynthesis through substrate specificity, enzyme kinetics, pathway branching, and neofunctionalization. Understanding the genetic regulation of key VOCs such as terpenoids, esters, and phenolics provides valuable insights into the molecular basis of fruit aroma. Gene stacking offers promising opportunities for breeding strategies aimed at enhancing fruit quality. By leveraging the benefits of different isoenzymes relevant with flower and fruit VOCs, breeders may potentially develop *Rosaceae* varieties with optimized aroma profiles that meet consumer preferences and industry demands better.

## Figures and Tables

**Figure 1 genes-15-01339-f001:**
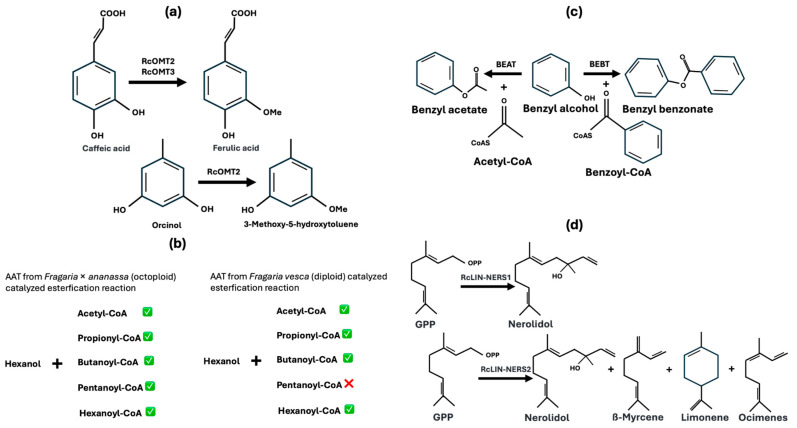
The variations in some VOCs production because of enzyme isoforms: (**a**) *Rosa chinensis* phenylpropanoid pathway, different caffeic acid O-methyltransferase (RcOMT) isoforms; (**b**) AAT isoforms from diploid strawberry and octoploid strawberry; (**c**) two members of BAHD family of acyltransferases acetyl-CoA:benzylalcohol O-acetyltransferase (BEAT) and benzoyl-CoA:benzyl alcohol benzoyl transferase (BEBT); (**d**), terpene synthase (TPS) isoforms (+)-(3S)-linalool nerolidol synthase 1&2 (RcLIN-NERS1&2).

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
