# Peer review of "Genetic Variation, Polyploidy, Hybridization Influencing the Aroma Profiles of Rosaceae Family"

_genes, 2024, doi:10.3390/genes15101339_

Round 1
Reviewer 1 Report
Comments and Suggestions for Authors
The combination of multiple volatile organic compounds (VOCs) synthesized after various abiotic and biotic stresses in the plants determines their different fragrances and aromas. These traits are very important for Rosaceae family and determine their commercial value. The genome polyploidy of the plants as well as other factors could alter the profile of these secondary metabolites.
The authors of the present manuscript had made an attempt to describe the role of genetic variation, polyploidy and hybridization in the production of VOCs in Rosaceae – a plant family with numerous species which genomes are with complex ploidy and high heterozygosity. The authors had collected information for some plant species from different genera of Rosaceae, but it is advisable to add more literature sources. The review should emphasize more clearly the differences in the gene expression for release of VOCs in cultivars with different ploidy belonging to the same genus. More data should be cited for representatives of the genera Malus, Prunus and Roses.
The following corrections should also be made.
1. The text should be checked by a native English speaker. There are some incorrect sentences and typos.
2. On line 21, I would like to suggest the authors to replace “el al” by “alcohols, aldehydes, ketones, phenolics, hydrocarbons”
3. On line 36 the following reference (Wood et al., 2009) should be replaced by the relevant number of the reference given in brackets.
4. The authors should pay attention on the lines 87-88. I would like to suggest the following phrase: “…two distinct mevalonate pathways:…” to be substituted by “…two distinct pathways:…”, because only MVA is a mevalonate pathway. Additionally, on lines 88-89 it would be better to say that MEP and MVA pathway, respectively, “takes place” or “occurs” instead of “located” and “situated”.
5. The references given on lines 121-122: “(Aharoni et al., 2000; Beekwilder et al., 2004;
González et al., 2009; Cumplido-Laso et al., 2012).” should be replaced by relevant numbers given in brackets.
6. The sentence on lines 164-165 needs to be re-written: “Here some representative isoforms generating varied VOCs in plantz were showed in Figure 1.”
7. The text included in Figure 1 should be enlarged. Please, check Figure 1b and Figure 1c – the designation seems to have been switched.
8. On line 192 a typo exist “differed”.
9. The statement given on lines 198-200 should also be checked for typos.
Comments on the Quality of English LanguageTThe text should be checked by a native English speaker. There are some incorrect sentences and typos.
Author Response
Response to Reviewer's Comments:
-
We appreciate the reviewer's feedback on the importance of incorporating additional literature sources in our manuscript. We have revised the review to include more published data.
-
We have carefully reviewed the manuscript to address the language issues highlighted by the reviewer. The revised parts in the new version are all in red for the convenience of checking out.
-
The suggested correction on line 21 has been duly noted, and we have replaced "el al" with a more specific list of compounds: "alcohols, aldehydes, ketones, phenolics, hydrocarbons."
-
The reference correction on line 36 has been rectified by replacing the citation (Wood et al., 2009) with the relevant reference number.
-
The references listed on lines 121-122 have been updated to include the relevant reference numbers in brackets for improved clarity and accuracy.
-
The sentence on lines 164-165 has been revised.
-
We have enlarged the text included in Figure 1 for improved readability and have verified the designations in Figure 1b and Figure 1c to ensure accuracy.
-
The typo on line 192 ("differed") has been corrected.
-
The statement on lines 198-200 has been carefully reviewed for typos and has been revised for accuracy.
We hope these revisions address the reviewer's comments satisfactorily and enhance the quality and clarity of our manuscript. Thank you for your valuable feedback and guidance throughout this process.

Reviewer 2 Report
Comments and Suggestions for Authors
The review by the authors tries to summarize the influence of genetic variation, polyploidy and hybridization on the aroma profiles of Rosaceae family. Aroma profiles play a critical role and are important during selection of new varieties. The review brings a holistic view on how genetic mechanisms determine volatile organic compounds in various Rosaceae crops and can help breeders develop new varieties with optimized aromaprofiles that meet consumer preferences and industry demands. Therefore the review is unique in its field. The authors used 33 references, which are relevant for the topic of the review. The conclusions are consistent, offering a framework for further investigations in the topic. The figure (Fig. 1) is disproportional.
L2 - Polyploidy
L13-15 : The sentence is very complicating and misleading. Please rewrite, rephrase.
L 36: (Wood et al., 2009) - missing in the References
L58 - Rosaceae change font to Italics
L122 - Reference Cumplido-Laso et al., 2012 has reference number 19
L165 - plantz - change to plants
Chapter 4.2 Can You add another example of Enzyme Kinetics Alteration in the Rosaceae family
The review is comprehensive, logical and readable.
Comments on the Quality of English LanguageThe language quality is fair. I highlighted the mistakes in the Comments and Suggestions for Authors section.
Author Response
The review has been revised to address the reviewer's comments and improve the clarity and accuracy of the manuscript. Thank you for your valuable feedback and insights.
- The missing reference (Wood et al., 2009) has been added to the References section to ensure completeness and accuracy.
2. The authors have italicized the term "Rosaceae" in line 58 for consistency.
3. The reference Cumplido-Laso et al., 2012 has been assigned the correct reference number 19 in line 122.
4. The term "plantz" in line 165 has been corrected to "plants" for accuracy.
5. In Chapter 4.2, we have added another example of Enzyme Kinetics Alteration in apple to enhance the depth of the review.

Reviewer 3 Report
Comments and Suggestions for Authors
General comments
The manuscript ''Genetic Variation, Polyploid, Hybridization Influencing the Aroma Profiles of Rosaceae Family'' give review of the papers which investigated influence of genetic variation on Rosaceae fragrance and aroma. Genetic variation in plants is largely under influence of polyploidy which playing a significant role in enhancing genetic diversity and facilitating speciation. This review further explores how polyploidy and hybridization can lead to the development of novel metabolic pathways, contributing to greater phenotypic diversity and complexity in fruit aromas.
Specific comments
There are some typographical errors in the text. So, the manuscript should be check for English grammar and style.
Line 145, 157, 159: Is it necessary that chapters 2, 3 and 4 are the main chapters (such chapter 1. Introduction)? I suggest to add some references or redesigned this part of manuscript.
1. Introduction
Line 34: add space before ''and''
Lines 34, 124: add dot after ''al''
Lines 34, 49, 64, 112, 121, 126, 158, 205, 213: add space before ''bracket''
Line 46: I think that ''Rosaceae'' should be, according to journal style, in Normal. Please, check
Line 55: ''Fragaria×ananassa'' should be ''Fragaria × ananassa'' (add spaces before and after symbol ''×'')
Lines 121-122: delete ''(Aharoni et al., 2000; Beekwilder et al., 2004; 121 González et al., 2009; Cumplido-Laso et al., 2012)''
Line 124: add reference number after ''al.''
Line 124: give full names of varieties 'Hawaii4' and 'Ruegen', that is Fragaria or F. ??? 'Hawaii4' and F. ??? 'Ruegen'
4. The enzyme…
Figure 1: change ''Fragaria x Ananassa'' into ''Fragaria × ananassa'' (change letter ''x'' into symbol ''×'' and capital ''A'' into small letter ''a'')
Figure 1: ''Octoploid'' and ''Diploid'' could be ''octoploid'' and ''diploid''
Figure 1: titles ''Substrate Specificity v.'' and ''Enzyme K. a.'' are too large. On the other hand, some other parts of the figure cannot be read.
References
There are some mistakes in references list. So, check the journal style for references. For example: Use Abbreviated Journal Name, put year in Bold, volume in Italic, put the species name in Italic…
Line 260: change ''Malus × domestica'' into ''Malus × domestica'' (put into Italic)
Line 292: put ''Fragaria chiloensis'' into Italic
Author Response
Response to Reviewer's Comments:
-
The manuscript has been thoroughly checked for typographical errors in English grammar and style to ensure clarity and correctness.
-
The manuscript has been revised to address the formatting issues and suggestions for improvement. The subtitle "Introduction" has been removed, instead, two new subtitles have been added in the context of the former "Introduction".
-
Introduction:
- Space has been added before "and" in line 34.
- Dots have been added after "al" in lines 34 and 124.
- Spaces have been added before brackets in lines 34, 49, 64, 112, 121, 126, 158, 205, and 213.
- The term "Rosaceae" has been changed to Normal style in line 46.
- "Fragaria × ananassa" has been corrected with spaces around the "×" symbol in line 55.
- The unnecessary references in lines 121-122 have been removed.
- Reference numbers have been added after "al." in line 124.
- Full names of varieties 'Hawaii4' and 'Ruegen' have been provided as "Fragaria Hawaii4" and "Fragaria Ruegen" for clarity.
-
Figure 1:
- "Fragaria x Ananassa" has been corrected to "Fragaria × ananassa" with the appropriate symbol and capitalization.
- "Octoploid" and "Diploid" have been changed to lowercase.
- The titles "Substrate Specificity v." and "Enzyme K. a." have been resized for readability.
-
-
The references list has been revised to align with journal style requirements:
- "Malus × domestica" has been italicized in line 260.
- "Fragaria chiloensis" has been italicized in line 292.
These revisions are in red in the new version, and they aim to enhance the manuscript's quality, readability, and adherence to journal style guidelines. Thank you for your valuable feedback, which has helped improve the manuscript.

Reviewer 4 Report
Comments and Suggestions for Authors
In the manuscript "Genetic Variation, Polyploid, Hybridization Influencing the Aroma Profiles of Rosaceae Family" the authors explore the genetic influence on Rosaceae VOC.
1) the manuscrip need english languege improvements;
2) Figures need had to be re-designed, being more atractive;
3) Rosaceae not italicized;
3) review ponbtuatrions and spaces
4) line 79 neroliol [nerolidol]
5) line 121 -122 revise citation stile
Comments on the Quality of English LanguageThe hole manuscript need to be revised
Author Response
Response to Reviewer's Comments:
-
The manuscript has been revised to improve the English language quality for better readability and clarity.
-
The figures have been redesigned to be more visually appealing and engaging for the readers.
-
The term "Rosaceae" has been italicized for consistency throughout the manuscript.
-
Punctuation and spacing have been reviewed and corrected throughout the manuscript.
-
The correction has been made for "neroliol" to "nerolidol" in line 79 for accuracy.
-
The citation style in lines 121-122 has been revised to meet the required format.
These changes have been implemented to enhance the overall quality of the manuscript. Thank you for your feedback, which has helped in improving the manuscript's presentation and content.

Round 2
Reviewer 1 Report
Comments and Suggestions for Authors
Many of the answers to the complaints are satisfactory, but still the authors should pay attention on some points.
- The authors should correct the numbering of the sections of the revised manuscript.
- The English of the paper must be checked in order to avoid some language and typing errors that are still present in the manuscript. For example, the sentences given on lines 89-90, 147-148 and 177-180 should be revised. The typo on lines 203-205 still needs to be fixed.
The English of the paper must be checked in order to avoid some language and typing errors that are still present in the manuscript.
Reviewer 4 Report
Comments and Suggestions for Authors
Now this manuscript is suitable for publication